# Dengue virus susceptibility in *Aedes aegypti* linked to natural cytochrome P450 promoter variants

Sarah H. Merkling [1,9], Elodie Couderc [1,2,9], Anna B. Crist[1], Stéphanie Dabo [1], Josquin Daron[1], Natapong Jupatanakul [1,3], Myriam Burckbuchler[4], Thomas Vial [1], Odile Sismeiro[5], Rachel Legendre [6], Adrien Pain [6], Hugo Varet [6], Davy Jiolle[1,7,8], Diego Ayala [7,8], Christophe Paupy [7,8], Eric Marois [4] & Louis Lambrechts [1] ✉

The mosquito *Aedes aegypti* is the primary vector for dengue virus (DENV), which infects millions of people annually. Variability in DENV susceptibility among wild *Ae. aegypti* populations is governed by genetic factors, but specific causal variants are unknown. Here, we identify a cytochrome P450-encoding gene (*CYP4G15*) whose genetic variants drive differences in DENV susceptibility in a natural *Ae. aegypti* population. *CYP4G15* is transiently upregulated in DENV-resistant midguts, while knockdown increases susceptibility, and transgenic overexpression enhances resistance. A naturally occurring 18-base-pair promoter deletion reduces *CYP4G15* expression and confers higher DENV susceptibility. The unexpected role of a cytochrome P450 in DENV susceptibility challenges the long-standing focus on canonical immune pathways and opens new avenues for understanding antiviral defense and DENV transmission in mosquitoes.

Dengue virus (DENV) is an emerging mosquito-borne pathogen causing hundreds of millions of infections globally each year[1]. The emergence and spread of DENV are tightly linked to its primary mosquito vector, *Aedes aegypti*[2]. Natural populations of *Ae. aegypti* show significant variability in their ability to become infected with DENV[3,4], which is largely governed by mosquito genetic factors[5,6]. Previous studies suggest that genes encoding digestive enzymes and canonical antiviral immune genes may contribute to this variation[6–11]. However, the specific causal gene variants underlying DENV susceptibility in *Ae. aegypti* remain unknown.

*Aedes aegypti* females acquire a DENV infection when they blood feed on a viremic human[12]. The mosquito midgut is the first organ to become infected[13]. The virus subsequently disseminates systemically, until it reaches the salivary glands, where it can be transmitted to the next human host. Once the infection is established, mosquitoes remain infected for the rest of their lifetime[13]. The probability of mosquito infection strongly depends on the infectious dose, that is, the concentration of virus particles in the bloodmeal[12]. At intermediate viremia levels, only a fraction of blood-fed mosquitoes become infected. This is due to both the stochastic nature of infection initiation by a small number of virions[14] and genetic variation in mosquito susceptibility[15]. Elucidating the *Ae. aegypti* genetic factors that influence the probability of midgut infection has been a long-standing quest because this knowledge could pave the way to novel control strategies to interrupt DENV transmission[16]. Here, we identify a cytochrome P450-encoding gene whose genetic variants drive differences in DENV susceptibility in a natural *Ae. aegypti* population.

[1]Institut Pasteur, Université Paris Cité, CNRS UMR2000, Insect-Virus Interactions Unit, Paris, France. [2]Sorbonne Université, Collège Doctoral, Paris, France. [3]National Center for Genetic Engineering and Biotechnology (BIOTEC), Khlong Luang District, Pathum Thani, Thailand. [4]Université de Strasbourg, CNRS UPR9022, INSERM U1257, Strasbourg, France. [5]Institut Pasteur, Digital Education Division, Education Department, Paris, France. [6]Institut Pasteur, Université Paris Cité, Bioinformatics and Biostatistics Hub, Paris, France. [7]MIVEGEC, Montpellier University, IRD, CNRS, Montpellier, France. [8]Centre Interdisciplinaire de Recherches Médicales de Franceville, Franceville, Gabon. [9]These authors contributed equally: Sarah H. Merkling, Elodie Couderc. ✉e-mail: louis.lambrechts@pasteur.fr

## Results and discussion

Genetic mapping through controlled crosses, comparing mosquitoes infected and uninfected after receiving the same infectious bloodmeal, is a valuable method for characterizing the genetic architecture of DENV susceptibility[6,17,18], however it often lacks gene-level resolution. To achieve gene-level resolution, we employed transcriptomic profiling to uncover specific genes whose expression is linked to infection status. Unlike conventional transcriptomic analyses comparing different mosquito genetic backgrounds (e.g., resistant vs. susceptible strains) or looking for virus-induced genes (e.g., infectious vs. mock bloodmeal), we compared the transcriptome of individual mosquitoes from the same population, that became infected or not after receiving the same infectious bloodmeal. We chose a wild-type *Ae. aegypti* strain originally from Bakoumba, Gabon (Fig. 1a) whose susceptibility to DENV type 1 (DENV-1) and type 3 (DENV-3) strains was previously characterized[19].

To identify the time frame during which the distinction between infected and uninfected mosquitoes becomes established after viral exposure, we monitored viral RNA levels and infectious titers over time in individual females that had received a bloodmeal containing a DENV-1 infectious dose expected to result in ~50% infection prevalence (Fig. 1b). On the day of the infectious bloodmeal, both viral RNA (quantified by RT-qPCR) and infectious particles (quantified by titration) were readily detected in all mosquitoes, however within the next 48 h viral RNA levels dropped to undetectable levels in about half of the mosquitoes, while infectious virus became undetectable in all mosquitoes (Fig. 1c, d). At later time points, ~50% of mosquitoes showed detectable viral RNA and infectious particles again, indicating that the infection outcome was determined within the first two days after the bloodmeal, a period when virus replication is restricted to the midgut tissue[20]. These observations are consistent with the "eclipse phase" reported in previous time-course experiments and bottleneck analyses showing that, shortly after *Ae. aegypti* mosquitoes ingest a DENV infectious bloodmeal, detectable virus levels drop transiently (often below the limit of detection) before reappearing later as newly produced virions[21,22]. To identify genes associated with DENV susceptibility or resistance, we compared the transcriptomes of individual midguts from DENV-positive and DENV-negative mosquitoes on day 1 and day 2 post infectious bloodmeal (Fig. S1 and Fig. S2). Only a small number of genes were differentially expressed at each time point (101 on day 1 and 83 on day 2) (Fig. 1e and Data S1), with no overlap between the two time points (Fig. 1f).

Of the 94 genes whose transcripts were enriched in DENV-exposed but uninfected midguts on day 1 or day 2 post bloodmeal, we selected 11 of them for functional validation based on their predicted function in immune or metabolic processes, either in *Aedes* mosquitoes or through their orthologs in *Anopheles* and *Drosophila* (Data S1). Only one candidate gene, *CYP4G15* (*AAEL006824*), encoding a cytochrome P450 enzyme, was successfully validated using RNAi-mediated gene silencing assays (Fig. 2a, b and Fig. S3). Cytochrome P450 monooxygenases are primarily known for their multiple roles in the metabolism of a wide range of substances, including drugs, toxins, and endogenous compounds such as hormones and fatty acids[23]. Suppressing *CYP4G15* expression increased DENV-1 infection prevalence from 44% to 77% (Fig. 2a, b and Fig. S4a). Conversely, transgenic overexpression of *CYP4G15* under the control of a systemic promoter (*Polyubiquitin*) caused a significant decrease (from 46% to 21%) in DENV-1 infection prevalence (Fig. 2c, d). *CYP4G15* overexpression also decreased DENV-1 infection prevalence (from 85% to 48%) using a higher bloodmeal titer (Fig. S4b, c). Additionally, *CYP4G15* overexpression resulted in a significant decrease (from 93% to 59%) in DENV-3 infection prevalence (Fig. 2e, f). In both gene silencing and overexpression experiments, *CYP4G15* affected the proportion of infected mosquitoes but had no detectable influence on viral RNA levels in infected mosquitoes on day 5 post infectious bloodmeal

(Fig. 2b, d, f), except in one experiment with a higher bloodmeal titer (Fig. S4c). Together, these results demonstrate that *CYP4G15* is an antiviral factor acting against both DENV-1 and DENV-3, that is transiently upregulated in the midgut of DENV-resistant mosquitoes and reduces the probability of midgut infection.

We observed substantial inter-individual variation in *CYP4G15* expression levels among wild-type mosquitoes from the Bakoumba strain (Fig. 2a, c, e; Fig. S4b). Consequently, we examined the possible contribution of genetic polymorphisms in *CYP4G15* to this variation. Sequencing the upstream region of the *CYP4G15* gene revealed a naturally occurring variant with an 18-base-pair (bp) deletion in the promoter sequence (Fig. 3a). Hereafter, we refer to the gene variant with the deletion as *CYP4G15*^Δ18 and to the gene variant without the deletion as *CYP4G15*^Δ0. We quantified *CYP4G15* expression in individual females of the Bakoumba strain 24 h after a non-infectious bloodmeal and genotyped their *CYP4G15* promoter. The estimated frequency of the *CYP4G15*^Δ18 variant was 20.8%, and accordingly mosquitoes homozygous for this variant were infrequent (~5%). We found that mosquitoes with one copy of the *CYP4G15*^Δ18 variant were associated with a significantly lower level of *CYP4G15* expression than mosquitoes homozygous for the *CYP4G15*^Δ0 variant (Fig. 3b). To determine the effect of the Δ18 deletion on gene expression, we generated transgenic reporter lines, in which *GFP* was placed under the control of the *CYP4G15*^Δ0 or *CYP4G15*^Δ18 promoters. We found that the 18-bp deletion alone is sufficient to drive differences in GFP expression at both the pupal and adult stages (Fig. 3c–e). Therefore, we discovered a naturally occurring deletion in the promoter region of *CYP4G15* that reduces its expression.

Given the link between *CYP4G15* expression and DENV susceptibility described above (Fig. 1), we hypothesized that the *CYP4G15*^Δ18 variant was associated with a higher DENV susceptibility relative to the *CYP4G15*^Δ0 variant. We leveraged a previous study in which mosquitoes from the Bakoumba strain were categorized as either resistant or susceptible to DENV-1 and DENV-3[19]. We genotyped the *CYP4G15* promoter region of these samples and found that the *CYP4G15*^Δ18 variant, which was present at an average frequency of 10.9% at the time of these experiments, was significantly more frequent in mosquitoes categorized as susceptible to either virus (Fig. 4a). To establish a direct link between *CYP4G15* variants and DENV susceptibility, we created two sub-strains of mosquitoes derived from the Bakoumba strain that were either homozygous for the *CYP4G15*^Δ18 variant or for the *CYP4G15*^Δ0 variant. Comparison of *CYP4G15* midgut expression kinetics between the sub-strains at 0, 1, 2, and 7 days post bloodmeal showed transient upregulation on day 1 only in the *CYP4G15*^Δ0 homozygous sub-strain, indicating a different midgut inducibility of the promoter variants (Fig. S5). However, our in silico analysis of the *CYP4G15* promoter did not identify any known transcription factor binding motifs that would be disrupted by the Δ18 deletion (Fig. S6a). In agreement with our hypothesis, dose-response experiments showed that mosquitoes from the *CYP4G15*^Δ18 homozygous sub-strain were significantly more susceptible to both DENV-1 and DENV-3 than mosquitoes from the *CYP4G15*^Δ0 homozygous sub-strain, which had a similar level of susceptibility as the parental Bakoumba strain (Fig. 4b; Table S1). We confirmed that one day after the infectious bloodmeal, the *CYP4G15*^Δ18 homozygous sub-strain had significantly lower *CYP4G15* expression levels than the *CYP4G15*^Δ0 homozygous sub-strain, and the parental Bakoumba strain (Fig. 4c). We verified that the difference in DENV susceptibility between the homozygous sub-strains was not influenced by the differential presence of known insect-specific viruses (Fig. S7). The *CYP4G15*^Δ18 homozygous sub-strain was also more susceptible to DENV-4 than the *CYP4G15*^Δ0 sub-strain, but no difference was observed for DENV-2, suggesting a degree of DENV type specificity (Fig. S8; Table S1). Finally, surveying publicly available genome sequences from wild *Ae. aegypti* specimens revealed the presence of the *CYP4G15*^Δ18 variant in several wild mosquito populations across West and Central

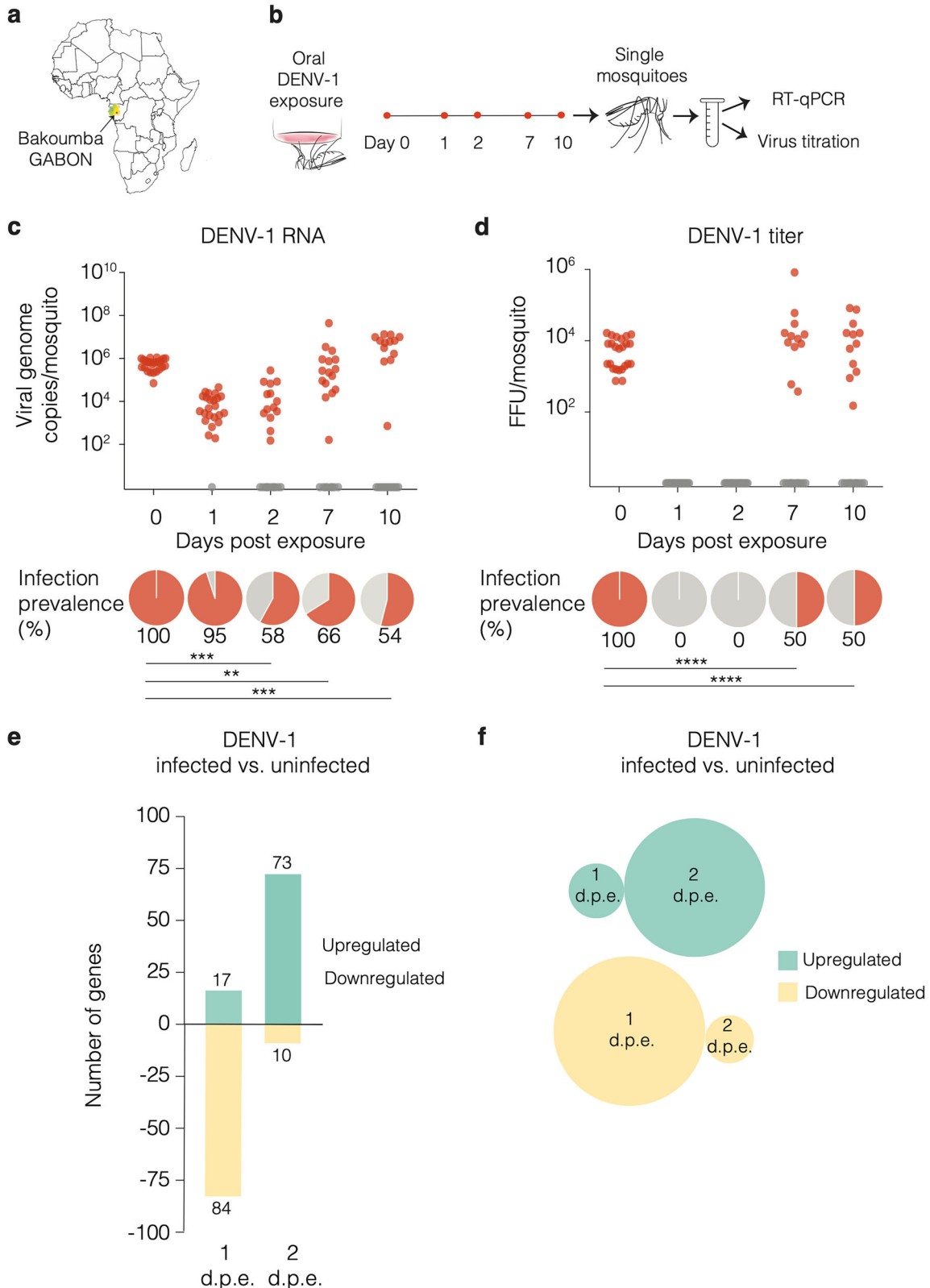

Africa (Fig. 4d). Specifically, the Δ18 deletion was detected in four *Ae. aegypti* populations from Senegal, Ghana, and Gabon, with frequencies ranging from 2.5% to 15.4%. This indicates that the *CYP4G15*[Δ18] variant occurs naturally in wild mosquito populations from West and Central Africa at frequencies similar to those observed in the Bakoumba strain.

We discovered a cytochrome P450-encoding gene of which naturally occurring variants drive differences in DENV susceptibility in *Ae.*

*aegypti*. The specific mode of action through which *CYP4G15* exerts its antiviral effect remains to be investigated. Enzymes of the *CYP4G* sub-family are known to catalyze the synthesis of cuticular hydrocarbons in insects[24,25]. These hydrocarbons facilitate desiccation resistance, modulate water loss, function as chemical signaling molecules, and play a role in the detoxification of xenobiotics. A previous study detected abundant transcripts of *CYP4G15* in *Ae. aegypti* oenocytes[26].

**Fig. 1 | Early transcriptomic analysis of individual midguts identifies *Ae. aegypti* genes associated with DENV-1 infection outcome. a** Map of the African continent (from Wikimedia Commons: https://fr.wikipedia.org/wiki/Fichier:Location_Gabon_AU_Africa.svg) showing the geographical origin (red dot) of the Bakoumba strain of *Ae. aegypti*. **b** Experimental scheme for the time-course analysis of infection outcome in *Ae. aegypti* mosquitoes from Bakoumba following an infectious bloodmeal containing $5 \times 10^6$ focus-forming units (FFU)/ml of DENV-1. Day 0 samples were collected just after the infectious bloodmeal. Viral RNA levels and infectious titers were determined on the same mosquito homogenates by RT-qPCR and virus titration, respectively. **c** Time course of DENV-1 RNA levels in single mosquitoes. The graph shows the abundance of viral RNA over time and the pie charts below represent the proportion of positive individuals ($n = 24$ mosquitoes per time point). Statistical significance of differences in infection prevalence was assessed relative to day 0 by chi-squared test (day 2: $p = 0.0004$; day 7: $p = 0.0019$; day 10: $p = 0.0002$) and shown in the figure (**$p < 0.01$; ***$p < 0.001$). **d** Time course of DENV-1 infectious titers in single mosquitoes. The graph shows the infectious titer over time and the pie charts below represent the proportion of positive individuals ($n = 24$ mosquitoes per time point). Statistical significance of differences in infection prevalence was assessed relative to day 0 by chi-squared test and shown in the figure (****$p < 0.0001$) except when prevalence was 0%, making the chi-squared test invalid. **e** Bar plot showing the number of differentially expressed genes between DENV-1-infected ($n = 8$) or uninfected midguts ($n = 8$) identified by RNA-seq 1 and 2 days post exposure (d.p.e.). Infection status of the samples was determined by RT-qPCR prior to RNA-seq (Fig. S1). A gene was considered differentially expressed when the fold change in transcript abundance was ≥2 and the adjusted $p$ value was ≤0.05 (Fig. S2c). **f** Venn diagrams showing the absence of overlap between differentially expressed genes at 1 and 2 d.p.e. Source data are provided as a Source Data file.

Interestingly, our transgenic reporter lines of *CYP4G15* promoter variants also displayed high levels of GFP expression that predominantly localized within pupal oenocytes (Fig. 3c). Oenocytes are ectodermic cells located in the fat body of insects, including mosquitoes, where they are involved in lipid metabolism and the biosynthesis of cuticular hydrocarbons[27]. It is possible that *CYP4G15* expression in oenocytes contributes to the antiviral effect observed in the midgut.

The natural geographic distribution of antiviral gene variants may contribute to explain the observed population-specific patterns of DENV susceptibility[28]. However, the evolutionary dynamics of susceptibility variants such as *CYP4G15*$^{\Delta18}$ are unlikely to be driven by antagonistic interactions between *Ae. aegypti* and DENV, because DENV infections have a low prevalence and a low fitness cost in wild mosquito populations, likely resulting in the absence of a DENV-driven selective pressure[29]. Pattern of linkage disequilibrium (LD) in wild-caught *Ae. aegypti* from Gabon indicate very limited LD between variants in the *CYP4G15* promoter and protein-coding regions (Fig. S6b). This suggests that the Δ18 deletion acts independently of SNPs in the protein-coding region of the gene.

The discovery of a cytochrome P450 involved in mosquito susceptibility to DENV infection challenges the prevailing dogma, which has primarily centered on canonical antiviral immune pathways such as Toll, IMD, JAK-STAT, and RNAi[30]. The antiviral role of *CYP4G15* is unexpected, given that cytochrome P450 enzymes are predominantly recognized for their roles in xenobiotic and endogenous compound metabolism, rather than antiviral defense[23]. Genes of the *CYP4G* subfamily are primarily involved in the synthesis of insect cuticular hydrocarbons[24,25], and also contribute to insecticide resistance in mosquitoes[31]. A previous study observed that the expression of *CYP4G15* was higher in an insecticide-resistant *Ae. aegypti* strain compared to a susceptible counterpart[32]. While most research on mosquito genes encoding cytochrome P450s has focused on insecticide resistance[33], our discovery suggests that insecticide resistance and virus susceptibility may be mechanistically linked, opening new perspectives for understanding mosquito-virus interactions. Potential mechanisms by which a cytochrome P450 enzyme might confer antiviral activity include modulating lipid metabolism crucial for the replication of enveloped viruses like DENV[34,35], or the production of reactive oxygen species that trigger immune responses and cellular defense mechanisms, or exert direct inhibitory effects on DENV[36,37]. Canonical immune pathways offer only a partial view of mosquito antiviral defense[38,39], and the recognition of non-canonical antiviral factors like *CYP4G15* presents exciting opportunities to further study DENV transmission dynamics and develop novel antiviral strategies in mosquitoes.

## Methods

### Ethics

Human blood samples to prepare mosquito artificial infectious bloodmeals were supplied by healthy adult volunteers at the ICAReB biobanking platform (BB-0033-00062/ICAReB platform/Institut Pasteur, Paris/BBMRI AO203/[BIORESOURCE]) of the Institut Pasteur in the CoSImmGen and Diagmicoll protocols, which had been approved by the French Ethical Committee Ile-de-France I. The Diagmicoll protocol was declared to the French Research Ministry under reference 343 DC 2008-68 COL 1. All adult subjects provided written informed consent. Genetic modification of *Ae. aegypti* was performed under authorizations number #7614, #3243 and #3912 from the French Ministry of Higher Education, Research, and Innovation. At IBMC in Strasbourg, mosquito husbandry involved bloodmeals on live mice that were approved by the CREMEAS Ethics committee and authorized by the French Ministry of Higher Education, Research, and Innovation under reference APAFIS #20562-2019050313288887v3.

### Mosquitoes

Experiments involved two wild-type (i.e., not genetically modified) strains and a preexisting genetically modified line of *Ae. aegypti*. A wild-type strain from Bakoumba, Gabon (referred to as the Bakoumba strain hereafter) was established from a natural population in 2014[19] and was used in this study between 8 to 27 generations of laboratory colonization. Genetically, the Bakoumba strain is predominantly assigned to the *Ae. aegypti formosus* (*Aaf*) subspecies (Fig. S9). A wild-type strain originally from Bangkok, Thailand (referred to as the Bangkok strain hereafter) was obtained by selecting wild-type individuals from the genetically modified MRA-863 strain[40] formerly distributed by BEI Resources (NIAID, NIH). The *Ae. aegypti* docking line X18A5 was created by excising the *Cp-Loqs2* transgene from a previously described transgenic line[41] via embryo microinjection of a Cre recombinase-expressing helper plasmid, leaving in the genome only a *piggyBac* insertion carrying an *att*P docking site, which was made homozygous. Mosquitoes were maintained as described previously at Institut Pasteur in Paris[42] and at IBMC in Strasbourg[43]. At Institut Pasteur, mosquitoes were reared at 28 °C ± 1 °C, under 70% ± 5% relative humidity and a 12/12-h light/dark cycle. At IBMC, mosquitoes were reared at 25–28 °C, under 75–80% relative humidity and a 14/10-h light/dark cycle. The larvae were fed a diet of fish food (Tetramin) and the adults were provided a 10% sucrose solution.

### Transgenesis plasmids

Plasmids for mosquito transgenesis were assembled by Golden Gate Cloning[44,45]. For this, each module to be included in the final assemblies (promoters, open reading frames, transcription terminators, fluorescence marker cassettes) was initially cloned with appropriate flanking *Bsa*I restriction sites in ampicillin-resistant vector pKSB- (Addgene ref. #62540). In a second step, relevant modules were assembled in the desired order into a final kanamycin-resistant transgenesis plasmid (*piggyBac* or *att*B docking plasmid) in a single *Bsa*I restriction-ligation reaction[45,46]. Two green fluorescent protein (GFP) reporter transgenes were designed under the control of the *CYP4G15* promoter with or without the Δ18 deletion. The promoter region without the Δ18 deletion (*Prom*$^{\Delta0}$) was amplified by PCR with mosquito genomic DNA from

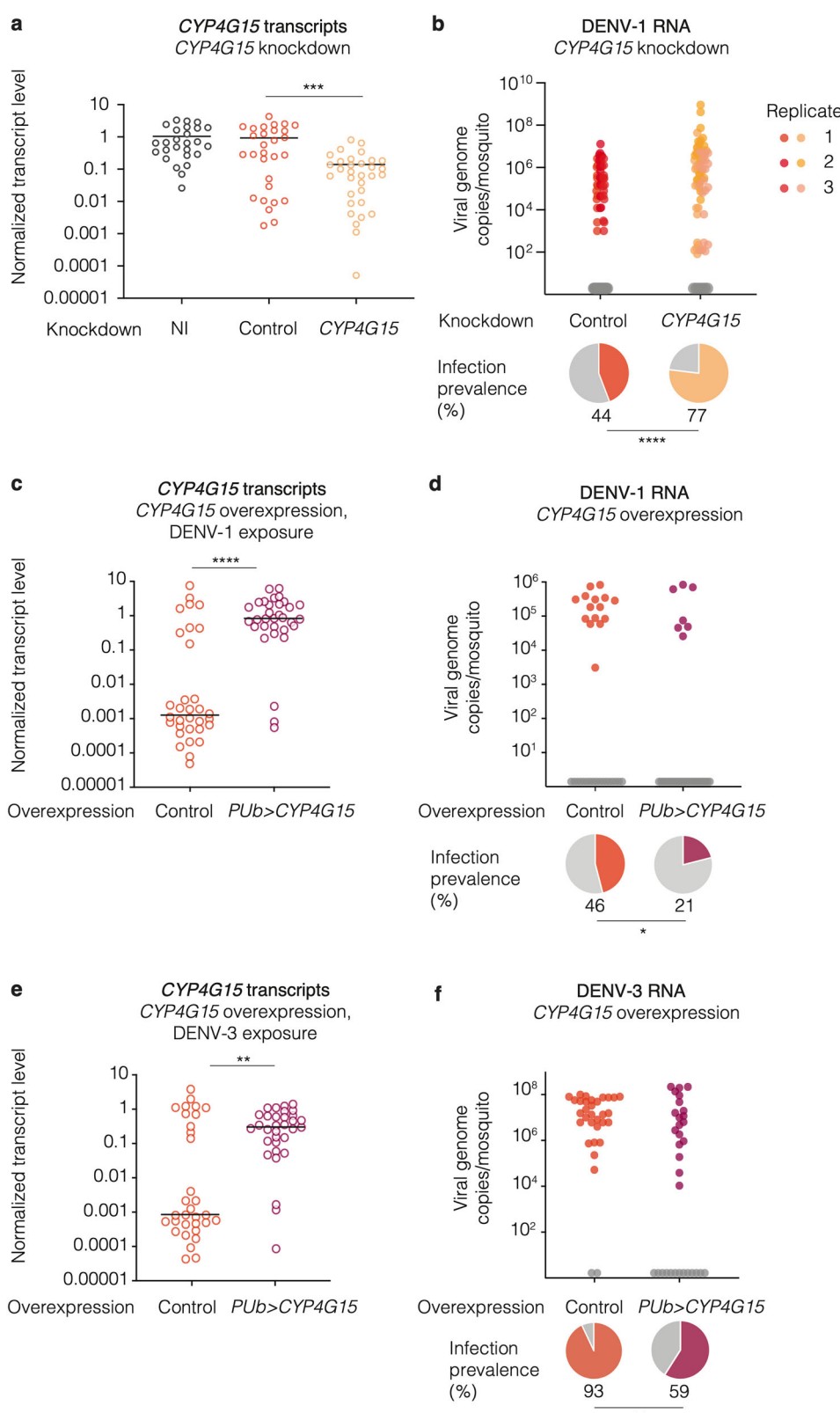

the Bakoumba strain using primers P1–P2 (Table S2). The promoter region with the Δ18 deletion (*Prom*^Δ18) was ordered as a synthetic DNA gBlock fragment (IDT DNA). The *CYP4G15* overexpression transgene was designed under the control of *Polyubiquitin* (*PUb*) promoter, which was amplified from the Bangkok strain with primers P3–P4 (Table S2). The *CYP4G15* open reading frame and transcription terminator region were amplified from the Bakoumba strain with primers

P5–P6 and P7–P8, respectively (Table S2). The Golden Gate Cloning-compatible destination vectors used were either a *piggyBac* backbone (Addgene ref. #173496) for the *PUb > CYP4G15* overexpression construct; or the *att*B docking plasmids pDSAT and pDSAR[45] (Addgene refs. #62290 and #62292) for the *Prom*^Δ0 > *GFP* and *Prom*^Δ18 > *GFP* reporter transgenes, respectively. The full annotated plasmid sequences are provided in Data S2.

**Fig. 2 | *CYP4G15* is an antiviral factor against DENV-1 and DENV-3. a** *CYP4G15* expression in whole mosquitoes upon gene silencing, 2 days after double-stranded RNA (dsRNA) injection targeting *CYP4G15* ($n = 31$) or *GFP* ($n = 29$) as a control. Non-injected (NI) mosquitoes ($n = 26$) were also included. Statistical significance of the pairwise differences was assessed by two-sided Mann–Whitney's test (*CYP4G15* vs. *GFP*: $p = 0.0001$). **b** DENV-1 RNA levels and infection prevalence in whole mosquitoes upon gene silencing of *CYP4G15* ($n = 91$) or *GFP* ($n = 95$) as a control. Viral RNA was quantified 5 days post DENV-1 exposure (7 days post dsRNA injection). The data presented are a combination of 3 experimental replicates. Viral RNA levels varied across replicates, represented by different color shades, while prevalence remained consistent across all replicates. Statistical significance of the overall difference in infection prevalence was assessed by chi-squared test ($p < 0.0001$). **c, e** *CYP4G15* expression in whole mosquitoes upon systemic *CYP4G15* overexpression and DENV-1 (**c**) or DENV-3 (**e**) exposure. Statistical significance of the pairwise differences ($n = 32$ mosquitoes per group) was assessed by two-sided Mann–Whitney's test

(DENV-1: $p < 0.0001$; DENV-3: $p = 0.0037$). **d, f** DENV-1 RNA levels and infection prevalence in whole mosquitoes upon systemic *CYP4G15* overexpression and DENV-1 (**d**) or DENV-3 (**f**) exposure. Statistical significance of the difference in infection prevalence ($n = 32$ mosquitoes per group) was assessed by chi-squared test (DENV-1: $p = 0.0353$; DENV-3: $p = 0.0012$). In **c–f**, *CYP4G15* was overexpressed transgenically under the control of a *Polyubiquitin* promoter (*PUb*) and mosquitoes were tested 5 days after DENV exposure. In **b, d, f**, the graph shows viral RNA levels, and the pie charts below represent the proportion of positive individuals. In **b–f**, the control line was the corresponding wild-type mosquito strain. Bloodmeal titers were $5 \times 10^6$ focus-forming units (FFU)/ml (**a, b**) and $10^7$ FFU/ml (**c, d**) of DENV-1, and $10^6$ FFU/ml (**e, f**) of DENV-3. In **a, c, e**, relative *CYP4G15* expression was calculated as $2^{-\Delta Ct}$, where $\Delta Ct = Ct_{CYP4G15} - Ct_{RP49}$, using the housekeeping gene *RP49* for normalization. In **a–f**, the horizontal bars represent the medians and statistically significant differences are shown (*$p < 0.05$; **$p < 0.01$; ***$p < 0.001$; ****$p < 0.0001$). Source data are provided as a Source Data file.

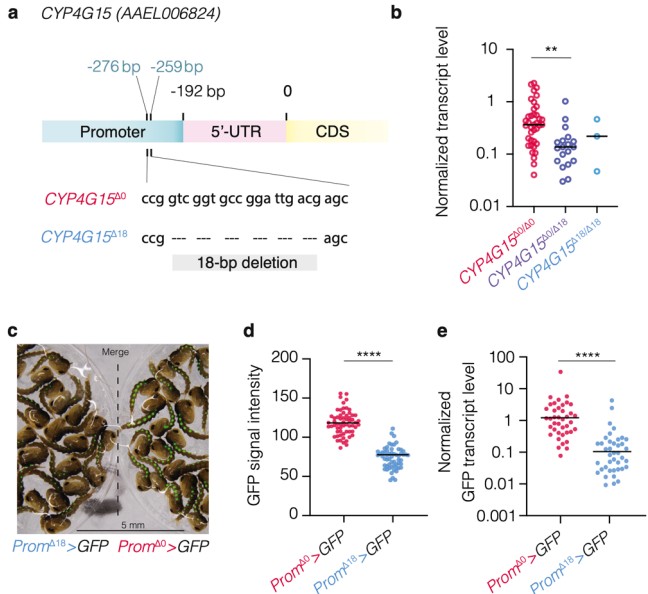

**Fig. 3 | Natural genetic variants of the *CYP4G15* promoter drive expression differences. a** Schematic representation of the two main genetic variants (*CYP4G15^Δ0^* and *CYP4G15^Δ18^*) of the *CYP4G15* promoter of mosquitoes from the Bakoumba strain, which differ by the presence/absence of an 18-bp deletion 259 bp upstream of the coding sequence (CDS) and 67 bp upstream of the 5′ untranslated region (5′-UTR). **b** *CYP4G15* expression in whole mosquitoes with different *CYP4G15* promoter genotypes ($n = 38$ *CYP4G15^Δ0/Δ0^* homozygotes, $n = 19$ *CYP4G15^Δ0/Δ18^* heterozygotes, and $n = 3$ *CYP4G15^Δ18/Δ18^* homozygotes), quantified by RT-qPCR. Statistical significance of the pairwise differences was assessed by two-sided Mann-Whitney's test (*CYP4G15^Δ0/Δ0^* vs. *CYP4G15^Δ0/Δ18^*: $p = 0.0002$). **c** Images of overlaid brightfield and GFP signals in transgenic pupae carrying a *GFP* transgene placed under the control of the *CYP4G15^Δ18^* (left) or *CYP4G15^Δ0^* (right) promoter, denoted as *Prom^Δ0^ > GFP* and *Prom^Δ18^ > GFP*, respectively. **d** Image quantification of GFP mean signal intensity in pupae from the transgenic *Prom^Δ0^ > GFP* ($n = 61$) and *Prom^Δ18^ > GFP* ($n = 58$) lines pictured in (**c**). Statistical significance of the difference was assessed by two-sided Mann-Whitney's test ($p < 0.0001$). **e** *GFP* expression in the whole bodies of adult female mosquitoes from the *Prom^Δ0^ >* GFP ($n = 40$) and *Prom^Δ18^ > GFP* ($n = 40$) reporter lines, quantified by RT-qPCR. Statistical significance of the difference was assessed by two-sided Mann-Whitney's test ($p < 0.0001$). In **b, e**, relative gene expression was calculated as $2^{-\Delta Ct}$, where $\Delta Ct = Ct_{Gene} - Ct_{RPS17}$, using the housekeeping gene *RPS17* for normalization. In **b, d, e**, the horizontal bars represent the medians, and statistically significant differences are shown (**$p < 0.01$; ****$p < 0.0001$). Source data are provided as a Source Data file.

## Transgenic mosquito lines

Transgenic mosquito lines were created from the Bakoumba strain (overexpression line) or from the X18A5 docking line (GFP reporter lines). Freshly laid eggs were aligned along a wet nitrocellulose

membrane as previously described[45] and injected with 400 ng/μl DNA in 0.5× phosphate-buffered saline (PBS) in the posterior pole, using quartz microcapillaries: 300 ng/μl transgenesis plasmid and 100 ng/μl helper plasmid encoding either *piggyBac* transposase[43] or PhiC31 integrase (Addgene ref. #183966). Generation 0 (G0) adult mosquitoes that emerged from the injected eggs were outcrossed with wild-type counterparts from the original strain. Transgenic larvae in the G1 progeny were identified by screening for fluorescence markers as previously described[43,47]. To create transgenic lines containing a single copy of the *PUb > CYP4G15* transgene, individual GFP-positive (GFP+) pupae were placed into Ø25 × 95 mm fly vials containing a small volume of water and sealed with a cotton plug (Flugs, Genesee Scientific) to isolate virgin adults. After adult emergence, individual GFP+ males were outcrossed to at least 10 wild-type females from the original strain. Subsequent individual GFP+ males were outcrossed with wild-type females until inheritance was approximately 50% with similar fluorescence levels among the GFP+ progeny. Lines where the transgene was sex-linked were discarded. Transgenic lines were established after three generations of outcrossing. Control "sister" lines were established by isolating the GFP-negative progeny from the final outcrossing generation[38]. In the case of the *Prom^Δ0^ > GFP* and *Prom^Δ18^ > GFP* transgenes, insertion events in the *att*P site of line X18A5 were amplified by outcrossing to wild-type mosquitoes and enriching for fluorescent individuals in subsequent generations. The *Prom^Δ0^ > GFP* and *Prom^Δ18^ > GFP* reporter lines were made homozygous by COPAS-selecting first-instar larvae carrying two copies of the transgene[48].

## Cells and virus isolates

C6/36 cells (derived from *Ae. albopictus*) were maintained in Leibovitz's L-15 medium (Life Technologies) supplemented with 10% fetal bovine serum (FBS, Life Technologies), 1% non-essential amino acids (Life Technologies), 2% tryptose phosphate broth (Gibco Thermo Fisher Scientific), 10 U/ml of penicillin (Gibco Thermo Fisher Scientific) and 10 μg/ml of streptomycin (Gibco Thermo Fisher Scientific) at 28 °C. DENV-1 isolate KDH0026A was originally derived in 2010 from the serum of a dengue patient in Kamphaeng Phet, Thailand[17]. DENV-2 isolate D2Gabon was originally derived in 2007 from the serum of a dengue patient in Libreville, Gabon[49]. DENV-3 isolate GA28-7 was originally derived in 2010 from the serum of a dengue patient in Moanda, Gabon[49]. DENV-4 isolate 63632 was originally derived in 1983 from the serum of a dengue patient in Senegal[50]. Informed consent of the patients was not necessary because viruses isolated in laboratory cell culture are no longer considered human samples. High-titer DENV stocks were prepared in C6/36 cells as previously described[42].

## Mosquito exposure to DENV

Mosquitoes were orally exposed to DENV as previously described[42]. Briefly, 5- to 7-day-old female mosquitoes were deprived of sucrose

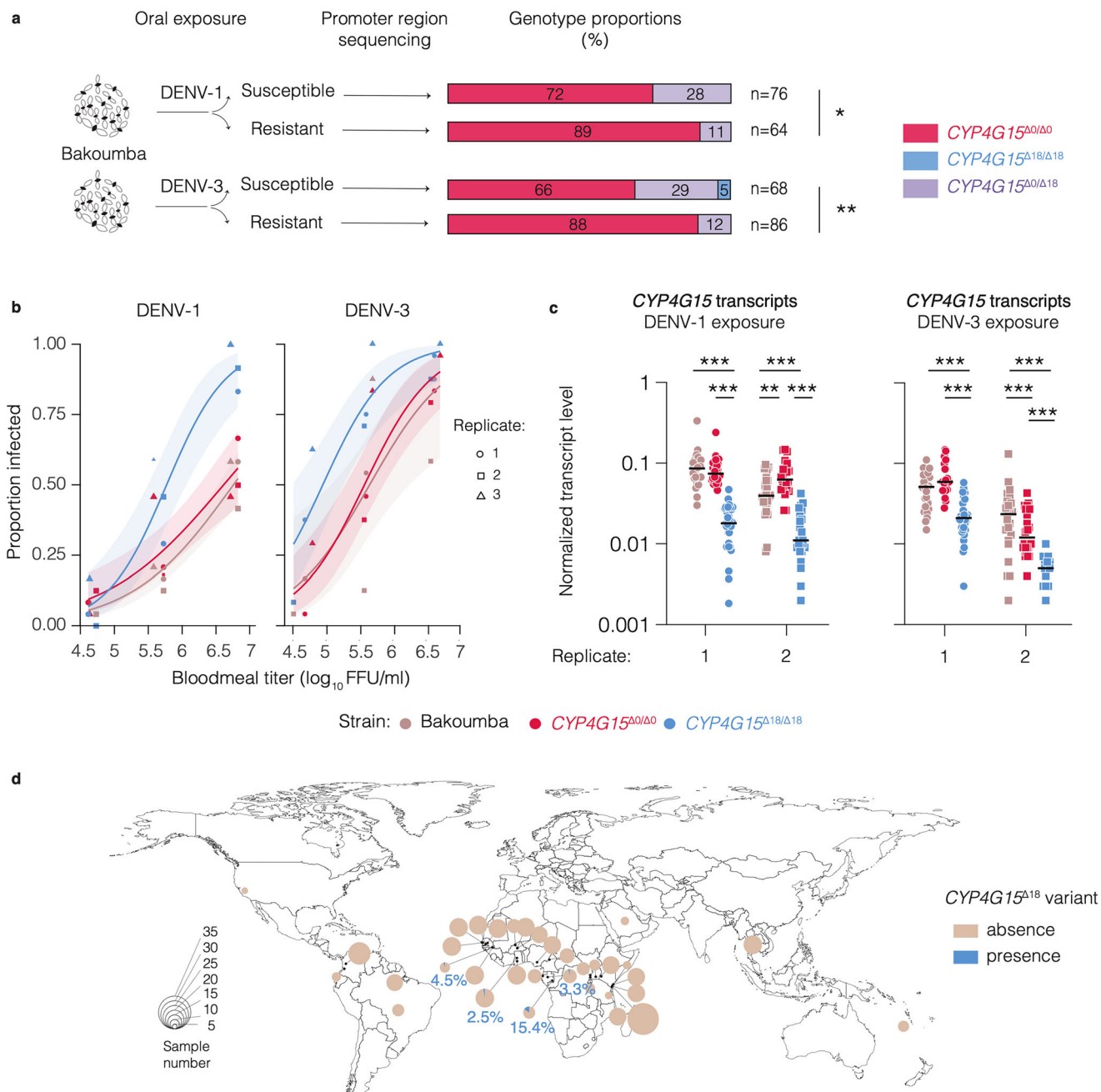

**Fig. 4 | *CYP4G15* promoter genotype contributes to natural variation in DENV susceptibility. a** Statistical association between *CYP4G15* promoter variants and phenotypic groups of mosquitoes from the Bakoumba strain categorized in a previous study[19] as either resistant or susceptible to DENV-1 and DENV-3, respectively. *CYP4G15* genotype was determined by Sanger sequencing of the promoter region. The total number of mosquitoes genotyped for each phenotypic group (*n*) is indicated next to the stacked bars. Statistical significance of the genotype-phenotype associations was assessed by Fisher's exact test and shown in the figure (**p* = 0.0189; ***p* = 0.0011). **b** Dose-response curves for DENV-1 (left) and DENV-3 (right) infection of the Bakoumba strain and two sub-strains homozygous for the *CYP4G15*^Δ18^ and *CYP4G15*^Δ0^ variants, respectively. In three experimental replicates, the proportion of mosquitoes positive for viral RNA 7 days post DENV exposure are shown as a function of the bloodmeal titer in $\log_{10}$-transformed focus-forming units (FFU)/ml. The size of the symbols is proportional to the sample size (*n* = 24 mosquitoes per group, except *n* = 22 for DENV-1 *CYP4G15*^Δ0^ medium dose in replicate 1 and DENV-1 *CYP4G15*^Δ18^ medium dose in replicate 3). The curves represent logistic fits of the data combined from the three replicates, with 95%

confidence intervals shown as shaded bands. The full statistical analysis of the dose-response curves is provided in Table S1. **c** *CYP4G15* expression in whole bodies of sub-strains homozygous for the *CYP4G15*^Δ18^ and *CYP4G15*^Δ0^ variants derived from the Bakoumba strain 1 day after DENV-1 (left) or DENV-3 (right) exposure (samples sizes from left to right: *n* = 28, *n* = 31, *n* = 32, *n* = 24, *n* = 24, *n* = 24, *n* = 28, *n* = 19, *n* = 30, *n* = 24, *n* = 24, *n* = 17). Relative gene expression was calculated as $2^{-\Delta Ct}$, where $\Delta Ct = Ct_{CYP4G15} - Ct_{RPS17}$, using the housekeeping gene *RPS17* for normalization. Data shown in **c** correspond to the highest (DENV-1) or intermediate (DENV-3) bloodmeal titers of experimental replicates 1 and 2 shown in (**b**). Statistical significance of the pairwise differences was assessed by two-sided Mann-Whitney's test and statistically significant differences are shown in the figure (***p* = 0.0034; ****p* < 0.0001). **d** Frequency of the *CYP4G15*^Δ18^ variant in wild *Ae. aegypti* populations worldwide (world map from the R package maps). Pie charts represent the proportion of individuals mosquitoes carrying at least one copy of the *CYP4G15*^Δ18^ variant detected in whole-genome sequences of populations from various geographical locations. The size of the circles represents the number of sequenced individuals per population. Source data are provided as a Source Data file.

solution 24 h before the infectious bloodmeal. Artificial infectious bloodmeals were prepared with human blood except in the experiments presented in Fig. 4b, c, which were performed with commercial rabbit blood (BCL) due to an interruption in human blood supply. Fresh whole blood was centrifuged for 15 min at 350$g$ to separate the erythrocytes from the plasma. The erythrocytes were washed 3 times in 1× PBS and centrifuged for 5 min at 1400$g$, before being resuspended in 1× PBS and supplemented with adenosine triphosphate at a final concentration of 10 mM. The infectious bloodmeal was a 2:1 mixture of washed erythrocytes and virus stock. It was offered to mosquitoes for 15 min through an artificial membrane-feeding system (Hemotek Ltd.) with pig intestine (Tom Press) as the membrane. Bloodmeal aliquots were collected prior to feeding to determine viral titer. Blood-fed females were incubated in 1-pint carton boxes under controlled conditions (28 °C ± 1 °C, 70% ± 5% relative humidity, 12/12-h light/dark cycle) with permanent access to a 10% sucrose solution.

### RNA extraction

To quantify viral RNA and individual gene expression levels, RNA was extracted from mosquito whole bodies using the NucleoSpin 96 RNA Core kit (Macherey-Nagel) following manufacturer's instructions. Shortly, samples were homogenized for 30 s at 6000 rotations per minute (rpm) in a Precellys 24 tissue homogenizer (Bertin Technologies) in 400 µl of RAV1 buffer with ~20 1-mm glass beads (BioSpec). Lysates were deposited on extraction columns and RNA eluted in 100 µl of RNase-free water. The protocol also included an on-column DNase treatment that was performed according to the manufacturer's instructions.

### Viral RNA quantification

DENV RNA was reverse transcribed and quantified using a TaqMan-based reverse transcriptase quantitative PCR (RT-qPCR) assay, using primers targeting a conserved region of DENV non-structural gene 5 (*NS5*) and a 6-FAM/BHQ-1 double-labeled probe (Table S2). Reactions were performed with the GoTaq Probe 1-Step RT-qPCR System (Promega) following the manufacturer's instructions. Standard curves of in vitro synthetized RNA dilutions were used to determine the absolute number of RNA copies per sample. Insect-specific virus RNA was reverse transcribed into complementary DNA (cDNA) using random hexameric primers and the M-MLV reverse transcriptase (Thermo Fisher Scientific) during 10 min at 25 °C, 50 min at 37 °C, and 15 min at 70 °C and quantified using the GoTaq BRYT-Green-based quantitative PCR assay (Promega) with specific primers for each insect-specific virus (Table S2), which were obtained from a previous study[51] except for *Aedes* anphevirus (Genbank accession number MH430665). Relative viral RNA levels were calculated as $2^{-\Delta Ct}$, where $\Delta Ct = Ct_{Virus} - Ct_{RPS17}$, using the *Ae. aegypti* ribosomal protein-coding gene *RPS17* (*AAEL004175*) for normalization.

### Qualitative DENV RNA detection

For dose-response experiments, DENV RNA was detected qualitatively using a two-step RT-PCR reaction targeting a conserved region of the DENV *NS5* gene as previously described[28]. Briefly, whole mosquito bodies were homogenized individually in custom buffer (Tris 10 mM, NaCl 50 mM, EDTA 1.27 mM, final pH adjusted to 9.2) supplemented with proteinase K (Eurobio Scientific) at a final concentration of 0.35 mg/ml. The homogenates were incubated for 5 min at 56 °C followed by 10 min at 98 °C. Total RNA was reverse transcribed into cDNA using random hexameric primers and the M-MLV reverse transcriptase (Thermo Fisher Scientific) during 10 min at 25 °C, 50 min at 37 °C, and 15 min at 70 °C. The cDNA was subsequently amplified using DreamTaq DNA polymerase (Thermo Fisher Scientific) and specific primer pairs for DENV-1 (P17–P18) and DENV-3 (P15–P16) (Table S2). The thermocycling program was 2 min at 95 °C, 35 cycles of 30 s at 95 °C, 30 s at 60 °C, and 30 s at 72 °C with a final extension step of 7 min at 72 °C. Amplicons were visualized by electrophoresis on 2% agarose gels.

### Gene expression measurement

Transcript abundance of individual genes was measured using a BRYT-Green-based RT-qPCR assay (GoTaq 1-Step RT-qPCR System, Promega), using gene-specific primers (Table S2) and following the manufacturer's instructions. Relative expression was calculated as $2^{-\Delta Ct}$, where $\Delta Ct = Ct_{Gene} - Ct_{RP49}$, using the *Ae. aegypti* ribosomal protein-coding gene *RP49* (*AAEL003396*) for normalization except in the experiments presented in Fig. 3b and Fig. 4c, which used *RPS17* (*AAEL004175*) instead. Quantification of *CYP4G15* expression was performed with primers P23-P24, except in the experiments presented in Fig. 3b and Fig. 4c, which required the use of degenerate primers P25-P26 to accommodate polymorphisms in the primer sequences of the *CYP4G15*$^{\Delta 0}$ and *CYP4G15*$^{\Delta 18}$ variants.

### DENV titration

Virus titration was performed by standard focus-forming assay as previously described[42]. In brief, a 96-well plate was seeded subconfluently with C6/36 cells, inoculated with the viral suspension, and covered with an overlay medium containing 1.6% carboxyl methylcellulose solution (Sigma). After 5 days of incubation at 28 °C, cells were fixed using 3.6% formaldehyde. Virus staining was performed using a primary mouse anti-DENV complex monoclonal antibody (MAB8705, Merck Millipore), and a secondary Alexa Fluor 488-conjugated goat anti-mouse antibody (Life Technologies). The infectious titer in focus-forming units (FFU)/ml was determined by counting infectious foci using a fluorescence microscope.

### RNA sequencing

The transcriptome of individual mosquito midguts was analyzed by RNA sequencing (RNA-seq). Mosquito midgut RNA was extracted with TRIzol (Life Technologies) as previously described[52]. The final RNA pellet was resuspended in 7 µl of RNase-free water and 0.5 µl were used for Nanodrop (Ozyme) quantification of RNA concentration. One µl of the RNA was diluted 10-fold and DENV RNA was quantified by RT-qPCR as described above to determine the infection status of each sample. At each time point (24 or 48 h post infectious bloodmeal), RNA samples from 8 infected and 8 uninfected midguts were selected based on their viral RNA quantity and quality (Fig. S1). The remaining 5.5 µl of RNA were treated with DNase I Ambion (Thermo Fisher Scientific) to eliminate potential DNA contaminants and run on Bioanalyzer using the Eukaryote Total RNA Nano Kit (Agilent) to accurately assess RNA quantity and quality. Following quality control, sequencing libraries were prepared using the TruSeq Stranded RNA LT Sample Prep kit set A (Illumina ref. #15032612) following the manufacturer's instructions. Before sequencing, library quality was confirmed on Bioanalyzer using the DNA High Sensitivity Kit (Agilent). Samples were normalized to 2 nM and multiplexed before being denatured by addition of 1 nM NaOH for 5 min at room temperature (20–25 °C). The multiplexed library was diluted to 10 pM and sequenced on a single-read flowcell v4 (65-bp reads) on a HiSeq 2500 instrument (Illumina). The raw RNA-seq data were deposited to NCBI Gene Expression Omnibus (GEO) under accession number GSE279387. The total read counts for each sample are shown in Fig. S2a.

### Transcriptomic analysis

RNA-seq reads with a quality score <30 were trimmed using Cutadapt version 1.11[53]. Passing-filter reads were mapped to annotated *Ae. aegypti* transcripts (AaegL5) using STAR version 2.5.0a[54] with default parameters. Non-annotated transcripts were identified by Trinity release version 2.4.0[55] from contigs >200 bp with >400 mapping reads. A functional annotation of novel transcripts was performed and

is available in GEO under accession number GSE279387. Reads mapping to both annotated and de novo transcripts were processed with featureCounts version 1.5.0-p3 from the Subreads package[56] to create a matrix of raw counts used for differential gene expression analysis. Count data were analyzed using R version 4.4.3[57] and the Bioconductor package edgeR version 4.4.2[58]. Genes with low expression (10,309 out of 22,715) were filtered out using the filterByExpr() function with min.count = 10. The normalization and dispersion estimation were performed using the default parameters. A generalized linear model was set in order to test for the differential expression between DENV-1-infected and uninfected midguts at each time point (1 and 2 days post exposure). For each comparison, raw $p$ values were adjusted for multiple testing according to the Benjamini and Hochberg procedure[59] and genes with a fold-change >2 and an adjusted $p$ value < 0.05 were considered differentially expressed. Principal component analysis and volcano plots are shown in Fig. S2b, c.

## Gene silencing

RNAi-mediated knockdown of *CYP4G15* expression was performed as previously described[52]. Briefly, double-stranded RNA (dsRNA) targeting *CYP4G15* was in vitro transcribed from T7 promoter-flanked PCR products using the MEGAscript RNAi kit (Life Technologies). To obtain the PCR products with a T7 promoter, a first PCR step was performed on genomic DNA extracted from mosquitoes of the Bakoumba strain using the Pat-Roman DNA extraction protocol as previously described[19]. The T7 sequence was then introduced during a second PCR step using T7 universal primers that hybridize to short GC-rich tags that were introduced to the PCR products in the first PCR (P35–P36; Table S2). Likewise, control dsRNA targeting *GFP* was synthesized using T7 promoter-flanked PCR products generated by amplifying a *GFP*-containing plasmid with T7-flanked PCR primers (P31–P32; Table S2). Newly synthesized dsRNA was resuspended in RNase-free water to reach a final concentration of 10 μg/μl. Five- to seven-day-old female mosquitoes from the Bakoumba strain were anesthetized on ice and injected intrathoracically with 140 nl dsRNA suspension using a Nanoject III apparatus (Drummond). After injection, mosquitoes were incubated for 2 days under standard insectary conditions before exposure to a DENV infectious bloodmeal. Specificity of *CYP4G15* silencing was verified by quantifying transcripts from the most closely related cytochrome P450-encoding gene, *CYP4G36* (*AAEL004054*).

## In situ GFP quantification

Transgenic pupae carrying the *GFP* reporter transgenes were imaged with a Nikon SMZ-18 fluorescence microscope. All images were taken with the same magnification and exposure time. Images were analyzed using Icy version 2.4.2.0[60]. GFP-positive patches were manually delimited, and mean GFP signal intensity was automatically computed using the Icy *region of interest* (ROI) analysis tool.

## CYP4G15 promoter genotyping

Genotyping of the *CYP4G15*$^{\Delta 0}$ and *CYP4G15*$^{\Delta 18}$ variants was performed by Sanger sequencing of the *CYP4G15* promoter region using primers P55-P56 for PCR amplification and primer P57 for sequencing (Table S2). Single mosquito legs were collected in 200 μl DNAzol DIRECT (DN131, Molecular Research Center Inc.). The legs were homogenized for 30 s at 6000 rpm in a Precellys 24 tissue homogenizer (Bertin Technologies), briefly centrifuged, and used within 20 min in PCR using DreamTaq DNA Polymerase (EP0701, Thermo Fisher Scientific) based on manufacturer's instructions. Approximately 0.6 μl of the DNA extract was mixed with 19 μl of DreamTaq PCR master mix. The PCR conditions consisted of 3 min of initial denaturation at 95 °C, 40 cycles of denaturation at 95 °C for 30 s, annealing at 58 °C for 15 s, extension at 72 °C for 45 s, followed by a final

extension step of 5 min at 72 °C. Amplicons of ~1000 bp were sent for commercial Sanger sequencing (Eurofins). Promoter genotype was determined based on the presence or absence of the Δ18 deletion in the sequencing chromatograms. Heterozygous individuals were associated with a recognizable pattern of double peaks starting at the site of the Δ18 deletion.

## CYP4G15 promoter analysis

The presence of transcription factor binding motifs in the *CYP4G15* promoter region was analyzed using the Motif Alignment and Search Tool (MAST) from the MEME Suite[61]. Motifs from the HOCOMOCO H12CORE collection[62] were queried within the 500-bp region upstream of the start codon (ATG) using default MAST parameters.

## Homozygous sub-strains of CYP4G15$^{\Delta 18}$ and CYP4G15$^{\Delta 0}$ variants

Sub-strains homozygous for the *CYP4G15*$^{\Delta 18}$ and *CYP4G15*$^{\Delta 0}$ variants were derived from the Bakoumba strain. Eggs from the Bakoumba strain were hatched and the larvae were reared as described above. To isolate adults and determine their *CYP4G15* genotype before mating, individual pupae were placed into Ø25 × 95 mm fly vials containing a small volume of water and sealed with a cotton plug (Flugs, Genesee Scientific). After adult emergence, a single leg was collected from cold-anesthetized adults for DNA extraction and genotyping as described above. Mosquitoes were then placed back into the vials to remain isolated and unmated until genotyping results were available. Mosquitoes homozygous for the *CYP4G15*$^{\Delta 18}$ and *CYP4G15*$^{\Delta 0}$ variants were sorted into separate cages and allowed to mate. The sub-strain homozygous for the *CYP4G15*$^{\Delta 0}$ variant was established from approximately 25 males and 25 females. The sub-strain homozygous for the *CYP4G15*$^{\Delta 18}$ variant was established from 3 males and approximately 30 females. The frequency of homozygous *CYP4G15*$^{\Delta 18}$ males was ~1% in the Bakoumba strain and recombinants were rare due to the sex linkage of the *CYP4G15* locus and the low recombination rate around the sex locus in males[63].

## Population survey of CYP4G15$^{\Delta 18}$ variant

Occurrence of the *CYP4G15*$^{\Delta 18}$ variant in natural *Ae. aegypti* populations was evaluated by surveying publicly available genomic data of wild *Ae. aegypti* specimens. Whole-genome sequences of 641 *Ae. aegypti* mosquitoes were retrieved from NCBI bioprojects PRJNA602495[64], PRJNA385349[65], PRJNA864744[66], and PRJNA943178[67]. Raw reads were trimmed using Cutadapt (-q 30 -m 50 --max-n 0)[53] and mapped to the AaegL5 reference genome assembly[18] using bwa-mem version 0.7.17[68] with default parameters. Individuals whose average genome sequencing depth was <8× (*n* = 8) were excluded from downstream analyses. After removing PCR duplicates with Picard tools (http://broadinstitute.github.io/picard), single-nucleotide polymorphisms (SNPs) were called using GATK HaplotypeCaller version 4.1.9.0[69]. SNPs within 5 kb of the *CYP4G15* locus were retained, and low-quality variants were filtered out if they failed any of the following criteria: QD < 5, FS > 60, or ReadPosRankSum <-8. Genotypes were included only if they had a genotype quality >20 and a sequencing depth ≥10×. The frequency of the *CYP4G15*$^{\Delta 18}$ variant was estimated using bcftools[70].

## Linkage disequilibrium at the CYP4G15 locus

Linkage disequilibrium (LD) patterns in the genomic region surrounding the *CYP4G15* gene were analyzed with LDBlockShow version 1.40[71] using whole-genome sequences from 43 wild *Ae. aegypti* specimens collected in Gabon[64].

## Subspecies assignment of the Bakoumba strain

The subspecies assignment of the Bakoumba strain was determined by identifying SNPs diagnostic for *Ae. aegypti formosus* (*Aaf*) and *Ae. aegypti aegypti* (*Aaa*), using the whole-genome sequences of individuals from Entebbe (Uganda) and Santarem (Brazil) as reference representatives, respectively[64]. To isolate the most discriminant

markers, 1,798 biallelic SNPs were selected under the criterion of near fixation (frequency >0.95) or near absence (frequency <0.05) in *Aaf*, coupled with the opposite allele frequencies in *Aaa*. Among these diagnostic 1798 SNPs for differentiation between the two subspecies, 154 were present in the pooled exome-sequencing data available for the Bakoumba strain[19]. The genetic background of the Bakoumba strain was determined by examining the allele frequency spectrum of these 154 SNPs (Fig. S9).

## Statistical analyses

Gene expression levels ($2^{-\Delta Ct}$ values), non-zero viral RNA levels, mean GFP signal intensities, and *GFP* transcript levels, were compared pairwise with two-sided Mann-Whitney's test, except for Fig. S5 where conditions were compared pairwise by one-way analysis of variance (ANOVA) after $\log_{10}$-transformation of the $2^{-\Delta Ct}$ values, followed by Tukey-Kramer's honest significance difference (HSD) test. Infection prevalence was analyzed by chi-squared non-parametric test. Genotype proportions were compared by two-sided Fisher's exact test. Dose-response curves were compared with multivariate logistic regression. Statistical analyses were performed in GraphPad Prism version 10.1.0 (www.graphpad.com) and JMP version 14.0.0 (www.jmp.com). The threshold for statistical significance was $p < 0.05$.

## Reporting summary

Further information on research design is available in the Nature Portfolio Reporting Summary linked to this article.

## Data availability

Raw RNA-seq data and functional annotation of novel transcripts are available from NCBI Gene Expression Omnibus under accession number GSE279387. Source data are provided with this paper.

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

## Acknowledgements

We thank Catherine Lallemand for assistance with mosquito rearing and the other members of the Lambrechts lab for their insights. We acknowledge the IBMC Insectarium facility (Institut de Biologie Moléculaire et Cellulaire, CNRS UAR1589, Strasbourg, France) for transgenic mosquito production. During the preparation of this work, the authors used ChatGPT-4 (OpenAI) to improve the readability and language of the manuscript. After using this tool, the authors reviewed and edited the content as needed and take full responsibility for the content of the publication. This work was supported by the French Government's Investissement d'Avenir program, Laboratoire d'Excellence Integrative Biology of Emerging Infectious Diseases (grant ANR-10-LABX-62-IBEID to L.L., S.H.M., and E.C.), Agence Nationale de la Recherche (grant ANR-18-CE35-0003-01 to EM and LL; grant ANR-17-ERC2-0016-01 to L.L.), and a Pasteur-Roux-Cantarini Fellowship (SHM). The funders had no role in

study design, data collection and analysis, decision to publish, or preparation of the paper.

## Author contributions

Conceptualization: S.H.M., E.M., L.L. Investigation: S.H.M., E.C., A.B.C., S.D., M.B., O.S., T.V., D.J., D.A., C.P., E.M. Data analysis: S.H.M., E.C., J.D., N.J., R.L., A.P., H.V., E.M., L.L. Visualization: S.H.M., E.C., J.D., L.L. Funding acquisition: S.H.M., E.C., E.M., L.L. Project administration: S.H.M., E.M., L.L. Supervision: S.H.M., E.M., L.L. Writing—original draft: S.H.M., L.L. Writing— review & editing: S.H.M., E.C., D.A., C.P., E.M., L.L.

## Competing interests

The authors declare no competing interests.
