## [Peer Review file · Nature Communications]

Dengue virus susceptibility in *Aedes aegypti* linked to natural cytochrome P450 promoter variants

Corresponding Author: Dr Louis Lambrechts

Version 0:

Reviewer comments:

Reviewer #1

(Remarks to the Author)

This study describes the characterization of a natural genetic factor associated with the susceptibility of the mosquito vector *Aedes aegypti* to dengue virus in Central Africa. Considering the major impact of this vector borne disease on global health and the remaining knowledge gaps about mosquito-virus genetic interactions, this study is of high interest for a broad research community. The manuscript is concise and well written. It presents an impressive scientific work spanning multiple nested experiments narrowing down the conclusions. Most results are supported by solid data and included appropriate statistical analyses. The interpretation is globally sound. This study is original as the mosquito genetic factor identified does not belong to gene families previously associated with canonical viral-immune pathways but affects a cytochrome P450 gene likely involved in cuticle hydrocarbons synthesis and which orthologs have been previously related to adaptation to insecticides.

As a preliminary step to the transcriptomic screening, the authors first confirm that mosquito females from a central African strain (Bakoumba strain) infected with a DENV-1 or DENV-2 show a decrease of viral RNA level and infectious particles within the 48h post infection, corresponding to a viral replication restricted to the midgut. Then, they compare the transcriptome of individual dissected midguts from DENV-1 negative versus positives mosquitoes at one- and two-days post infection. This transcriptomic screening provides a set of differentially transcribed genes at each time point. From this gene list (186 genes), the authors focus on the cytochrome P450 gene *Cyp4g15* (accession number AAEL006824) which is overtranscribed one-day post infection in DENV-1 negative mosquitoes. The rest of the study aims at validating the importance of this gene in DENV susceptibility. First, the authors used transient RNAi to knock down *CYP4G15* expression and show that this leads to a significant increased DENV-1 prevalence. Reciprocally, transgenic overexpression under the control of a systemic promoter confirms that the over-expression of this P450 leads to a decreased DENV-1 and DENV-3 prevalence. Sequencing the *cyp4g15* promoter of mosquitoes from the Bakoumba strain revealed the presence of a 18bp deletion associated with its downregulation. The importance of this promoter variant in gene transcription is then validated using transgenic reporter lines. An association between this promoter variant and DENV-1 and DENV-3 susceptibility is then shown using individual mosquitoes and further confirmed using lab-made homozygous sub-lines. Finally, using publicly available genome sequences from worldwide populations, the occurrence of this rare P450 promoter variant is confirmed in a few populations from Central to West Africa.

First, I would like to congratulate authors for this interesting and extensive study which is convincing about the role of *cyp4g15* in the susceptibility of *Ae. aegypti* to DENV infection. In this regard, I believe that the few comments below should help improving the overall quality and robustness of the study.

1) The rationale of picking this particular P450 from the whole RNA-seq dataset is not obvious and should be made clearer to the reader. More precisely, it is unclear why a preference was made to genes being overexpressed in DENV-1 negative mosquitoes as compared to undertranscribed genes and why differentially transcribed from the other timepoint were barely considered. A few more words justifying this choice would clearly improve the manuscript. In addition, providing the 'gene description field' for all differentially transcribed genes at each time point in data table S2 would improve the interest of the community for this original dataset. Having said this, it looks like that a few individuals used for RNA-seq show extremely high normalized read counts as compared to others (sometime >100 fold) rising the concern of a potential bias in this data set. This is particularly worrying as a relatively low number of individuals were used for each condition (n=8) making these

individuals strongly impacting the mean fold changes between conditions. This might also explain the unbalance observed between over- and undertranscribed genes and possibly the absence of overlap between the two timepoints. In order to reassure the reader about this, I suggest confirming if these individuals are not true outliers by performing a PCA on all individuals from each timepoint using normalized read counts from all detected genes. If those individuals still appear as outliers, this will confirm that something is wrong in this RNA-seq dataset which might be related to unspecific midgut dissection or sample preparation problems. In any case, providing a global overview of the dataset (i.e. read mapping and filtering statistics, PCA on individuals, volcano plots, ...) should allow improving reader confidence in this dataset.

2) It is not clear why different DENV serotypes were used through the study which might interrogate the reader about what is shown and what is not. In addition, as other DENV serotypes are circulating in Africa, examining if the studied P450 variant also impacts DENV-2 and DENV-4 infection would be of interest as it may inform about the specificity of this P450-related DENV susceptibility factor.

3) It is stated that *cyp4g15* is 'transiently' regulated in the midgut of DENV-resistant mosquitoes under the control of the studied promoter variant. Though the transient regulation of such P450 gene is likely, I did not see much information about its regulation according to blood meal (infected or not), tissues or life stages in the present study. If such promoter region contributes to the transient regulation of *cyp4g15* following a DENV infectious blood meal, this may deserve to be supported by experimental data. If this promoter deletion truly contributes to the 'transient' regulation of this P450, it might be worth looking for transcription factor binding sites matching this 18 bp sequence. By the way, I suggest that the full *cyp4g15* promoter sequences obtained from the current study should be made available to the reader (suppl. file showing sequence alignment?).

Minor comments

- I suggest to change the title as follows: Dengue susceptibility in *Aedes aegypti* is linked to a natural cytochrome P450 promoter variant.

- It has been shown that *AeCYP4G15* (AAEL006824) is preferentially expressed in mosquito oenocytes (see Martins et al. 2011, doi:10.1590/s0074-02762011000300009). In the present study, the role of this gene in DENV susceptibility is thought to be associated to its transient regulation in the midgut of adult females. How can the reader be sure that the effect of this P450 variant is restricted to the midgut and not more related to other tissues?

- While getting through RNA-seq methods, I noticed that no minimum 'normalized read count' threshold was applied to the dataset, leading to retain genes showing very low read count across samples (which usually often leads to overestimating fold changes through random draw effect). I understand that applying such filter will likely remove some differentially expressed genes but this may also help consolidating this tissue-specific RNA-seq dataset.

- Two distinct *Ae. aegypti* forms appears to coexist in Central Africa (*aegypti aegypti* and *aegypti formosus*, see Crawford et al. doi.org/10.1101/2024.07.23.604830) but it is not said which form is represented in the Bakoumba strain. In this concern, I suggest providing more information about the origin and the biological nature of Bakoumba strain in the method section. In addition, it would be informative for the reader to know the *aegypti/formosus* status of field collected individuals genotyped for the *cyp4g15* promoter deletion in fig4d.

Line 146: 'is not an artefact of lab colonization'. Not sure to understand this... Although laboratory colonization can lead to strong demographic effects, there is barely no chance that colonization would lead to the de novo emergence of such mutation...

Line 148: change to 'a naturally occurring variant'

Lines 151-153: Not sure to understand this. I believe that the low frequency of this variant in natural populations and its relative narrow geographic range may also support a significant fitness cost impairing its spread throughout Africa?

Lines 165-166: I am not sure this example is pertinent considering that *cyp4g15* does not typically show the profile of a xenobiotic-metabolising P450 such as CYP6Ps. Indeed, this gene does not belong to classical P450 subfamilies associated with xenobiotic metabolism in mosquitoes, is highly conserved among species (slow evolution) and is preferentially expressed in oenocytes. Though this gene has been associated to insecticide resistance (Grigoraki et al 2020 Elife, Despres et al Biol letters 2014, Reid et al 2014 J Med Entomol) its role in CHC synthesis is far more credible.

Lines 323-324: Degenerate primers were used for comparing the expression of *cyp4g15* between individuals and pure lines carrying or not the promoter deletion to 'accommodate for polymorphisms'. But then I wonder how such polymorphisms may affect the *cyp4g15* transcription data presented early on in the manuscript.

Line 357: in Data S2, you may want to indicate the inferred function for novel transcripts.

Lines 430-431: Such polymorphism dataset may allow investigating if other coding variants are associated with the 18bp promoter deletion...

Fig4d. Please provide allele frequencies instead of occurrence frequencies.

- All figures. Please change 'normalized expression' into 'normalized transcript level'.

- Whole manuscript: the term 'wild-type strain' is confusing. Change to 'field-derived strain'.

Reviewer #2

(Remarks to the Author)

This is a novel approach for mapping genes of interest in virus susceptibility – focusing on comparisons between individuals that vary within a population. The authors survey gene expression on days 1 and 2 post feeding in the midgut and find only a few number of genes associated with differences in infection status and that they do not overlap. This suggests distinct mechanisms for controlling infection phenotypes recognizable at different DPI. They follow up with some robust functional work focused on a single cytochrome P450 gene and identify an upstream deletion associated with differences in expression. They also make homozygous lines to look at susceptibility to 2 DENV serotypes and explore the presence of the variant in published sequences. P450's have not traditionally been associated with antiviral activity. This is a nice contribution to the field.

The writing is clear and the experimental approaches robust.

I just have some suggestions that will not require any further experimental work.

What were the full list of 18 genes you found at DPI 1? What are their functions? A summary of these genes should be included (not just relegated to the Supp). Why did you focus on this single gene for this study? Please add the reasoning to the text (couldn't knock down expression of the others?, functions uninteresting?). Wanting to save the other genes for other papers does not seem like a good enough reason as this paper is not very long. They should at least be discussed. Functional work could follow in other papers.

Does this variant occur outside Africa? Or does this suggest a single local evolutionary event + spread. Finding it more frequently in a single copy only could indicate fitness issues. I would remove the statement about the variant unlikely causing fitness effects.

Reviewer #3

(Remarks to the Author)

Merkling et al. investigated variation in DENV infection within an *Aedes aegypti* population by comparing the transcriptomes of DENV-infected and uninfected mosquito midguts at 1 and 2 days post-infection. Among the differentially expressed genes, they focused on a p450 gene, CYP4G15. The expression of this gene exhibited variation within the population. When the gene was silenced, the proportion of infected mosquitoes increased, but not virus replication. Conversely, overexpression of the gene resulted in a reduced proportion of DENV-infected mosquitoes using DENV-1 and DENV-3. Comparison at the genomic level revealed variation in the promoter region of CYP4G15, with those possessing an 18-bp deletion showing significantly lower CYP4G15 expression. This deletion was also associated with a higher proportion of infected mosquitoes. The results of this study are very interesting, revealing a mosquito genetic factor involved in variation in their susceptibility to DENV. This is the first time a p450 gene is shown to be involved in resistance to DENV. The manuscript is also well-written. I have only minor comments.

Line 85-86: what other genes were differentially expressed, and why was CYP4G15 selected among the rest?

Line 169: A quick search on PubMed (DOI: 10.4269/ajtmh.18-0607) showed that, in at least one instance, upregulation of CYP4G15, among other things, is associated with insecticide resistance in *Aedes aegypti*, which might be worth mentioning.

Lines 270-271: providing the g force for the speeds rather than rpm is better.

Line 284, 337: RNase

Line 360: to create a matrix

DATA S2 and line 82: there are only 7 genes listed in DATA S2.

Version 1:

Reviewer comments:

Reviewer #1

(Remarks to the Author)

First, I would like to acknowledge the additional work performed by authors to answer reviewers' comments. Overall, I believe that the additional data provided and the modifications made to the manuscript improved its overall quality, making it suitable for publication in Nature Communications.

Below are two last comments that may help authors to further improve the manuscript.

1) The additional data provided suggest that the Bakoumba strain is a mixture of *Ae. aegypti aegypti* and *Ae. aegypti formosus* with the later genetic background being predominant.

Such finding raises the question of the possible association of the CYP4G15 promoter deletion variant to a specific molecular form, which can be related to its low observed frequency in the field and its narrow geographical distribution in

Fig4d. In this view, it would be interesting to know the molecular form of the field caught specimens genotyped for the deletion in Fig4d. In other words, is there any association between the presence of the CYP4G15 deletion variant and a given molecular form?

Similarly, considering that both forms are interfertile but that introgression will lead to chromosomal regions being closer to one or the other molecular form, it would be interesting to compare the genomic sequence surrounding the CYP4G15 locus between 'deletion' and 'wild type' homozygous lines and see if they rather match to one or the other molecular form.

I believe that the possible association of this molecular variant to a particular *Ae. aegypti* molecular form of interest for the reader as it may provide clues about its origin, its dynamics in the field and its association with DENV transmission. In this view I think such questions are worth to be discussed.

2) minor comment,

I understand the authors' point of view about not using the singular form 'a naturally occurring variant' though I believe that the term "variant" implies that there are at least two distinct forms coexisting in natural populations.

Below are the point-by-point responses (in blue font) to the Reviewers' comments (in black font). The line numbers refer to the clean version of the revised manuscript.

Reviewer #1 (Remarks to the Author):

This study describes the characterization of a natural genetic factor associated with the susceptibility of the mosquito vector *Aedes aegypti* to dengue virus in Central Africa. Considering the major impact of this vector borne disease on global health and the remaining knowledge gaps about mosquito-virus genetic interactions, this study is of high interest for a broad research community. The manuscript is concise and well written. It presents an impressive scientific work spanning multiple nested experiments narrowing down the conclusions. Most results are supported by solid data and included appropriate statistical analyses. The interpretation is globally sound. This study is original as the mosquito genetic factor identified does not belong to gene families previously associated with canonical viral-immune pathways but affects a cytochrome P450 gene likely involved in cuticle hydrocarbons synthesis and which orthologs have been previously related to adaptation to insecticides.

As a preliminary step to the transcriptomic screening, the authors first confirm that mosquito females from a central African strain (Bakoumba strain) infected with a DENV-1 or DENV-2 show a decrease of viral RNA level and infectious particles within the 48h post infection, corresponding to a viral replication restricted to the midgut. Then, they compare the transcriptome of individual dissected midguts from DENV-1 negative versus positives mosquitoes at one- and two-days post infection. This transcriptomic screening provides a set of differentially transcribed genes at each time point. From this gene list (186 genes), the authors focus on the cytochrome P450 gene *Cyp4g15* (accession number AAEL006824) which is overtranscribed one-day post infection in DENV-1 negative mosquitoes. The rest of the study aims at validating the importance of this gene in DENV susceptibility. First, the authors used transient RNAi to knock down CYP4G15 expression and show that this leads to a significant increased DENV-1 prevalence. Reciprocally, transgenic overexpression under the control of a systemic promoter confirms that the over-expression of this P450 leads to a decreased DENV-1 and DENV-3 prevalence. Sequencing the *cyp4g15* promoter of mosquitoes from the Bakoumba strain revealed the presence of a 18bp deletion associated with its downregulation. The importance of this promoter variant in gene transcription is then validated using transgenic reporter lines. An association between this promoter variant and DENV-1 and DENV-3 susceptibility is then shown using individual mosquitoes and further confirmed using lab-made homozygous sub-lines. Finally, using publicly available genome sequences from worldwide populations, the occurrence of this rare P450 promoter variant is confirmed in a few populations from Central to West Africa.

First, I would like to congratulate authors for this interesting and extensive study which is convincing about the role of *cyp4g15* in the susceptibility of *Ae. aegypti* to DENV infection. In this regard, I believe that the few comments below should help improving the overall quality and robustness of the study.

1) The rationale of picking this particular P450 from the whole RNA-seq dataset is not obvious and should be made clearer to the reader. More precisely, it is unclear why a preference was made to genes being overexpressed in DENV-1 negative mosquitoes as compared to undertranscribed genes and why differentially transcribed from the other timepoint were barely considered. A few more words justifying this choice would clearly improve the manuscript. In addition, providing the 'gene description field' for all differentially transcribed genes at each time point in data table S2 would improve the interest of the community for this original dataset. Having said this, it looks like that a few individuals used for RNA-seq show extremely high normalized read counts as compared to others (sometime >100 fold) rising the concern of a potential bias in this data set. This is particularly worrying as a relatively low number of individuals were used for each condition (n=8) making these individuals strongly impacting the mean fold changes between conditions. This might also explain the unbalance observed between over- and undertranscribed genes and possibly the absence of overlap between the two timepoints. In order to reassure the reader about this, I suggest confirming if these individuals are not true outliers by performing a PCA on all individuals from each timepoint using normalized read counts from all detected genes. If those individuals still appear as outliers, this will confirm that something is wrong in this RNA-seq dataset which might be related to unspecific midgut dissection or sample preparation problems. In any case, providing a global overview of the dataset (i.e. read mapping and filtering statistics, PCA on individuals, volcano plots, ...) should allow improving reader confidence in this dataset.

Response: We have provided a justification to the choice of *CYP4G15* as the primary focus of the functional follow-up experiments on lines 87-92. In short, we selected 11 targets for functional testing, prioritizing genes that had a predicted or confirmed link to immunity or metabolism. Amongst these, only *CYP4G15* significantly impacted dengue virus RNA levels in gene silencing assays. We have included a new Figure S3 presenting the results of the gene silencing assays for the other 10 shortlisted candidate genes, also listed in Data S1 (formerly Data S2).

Following the Reviewer's suggestions, we have updated Data S1 to include a description of gene function and provided a graphical overview of the RNA-seq dataset (total read counts, PCA, and volcano plots) in a new Figure S2. The barplot of total reads count per sample (Figure S2a) shows moderate variation among samples, which is accounted for by the normalization procedure. The absence of major bias in the RNA-seq dataset was

confirmed by the lack of consistent outliers in the PCA (Figure S2b).

2) It is not clear why different DENV serotypes were used through the study which might interrogate the reader about what is shown and what is not. In addition, as other DENV serotypes are circulating in Africa, examining if the studied P450 variant also impacts DENV-2 and DENV-4 infection would be of interest as it may inform about the specificity of this P450-related DENV susceptibility factor.

Response: We initially focused on DENV-1 and DENV-3 by consistency with our previous study of the Bakoumba strain (Dickson et al. 2020). To address the Reviewer's comment, we performed a new dose-response experiment to compare DENV-2 and DENV-4 susceptibility between the *CYP4G15*^{Δ18} and the *CYP4G15*^{Δ0} homozygous sub-strains. The results are presented in a new Figure S8 and revised Table S1. We found that the *CYP4G15*^{Δ18} homozygous sub-strain is more susceptible to DENV-4 than the *CYP4G15*^{Δ0} sub-strain, however no difference was observed for DENV-2, suggesting a degree of DENV type specificity. The manuscript was updated accordingly on lines 148-151.

3) It is stated that *cyp4g15* is 'transiently' regulated in the midgut of DENV-resistant mosquitoes under the control of the studied promoter variant. Though the transient regulation of such P450 gene is likely, I did not see much information about its regulation according to blood meal (infected or not), tissues or life stages in the present study. If such promoter region contributes to the transient regulation of *cyp4g15* following a DENV infectious blood meal, this may deserve to be supported by experimental data. If this promoter deletion truly contributes to the 'transient' regulation of this P450, it might be worth looking for transcription factor binding sites matching this 18 bp sequence. By the way, I suggest that the full *cyp4g15* promoter sequences obtained from the current study should be made available to the reader (suppl. file showing sequence alignment?).

Response: We performed a new time-course experiment to further characterize the kinetics of *CYP4G15* expression in the midgut according to the *CYP4G15* promoter genotype. We measured *CYP4G15* expression in the *CYP4G15*^{Δ18} and the *CYP4G15*^{Δ0} homozygous sub-strains, 0, 1, 2, and 7 days after ingesting a bloodmeal. The results are presented in a new Figure S5. We found that *CYP4G15* is transiently upregulated in the midgut at 1 day post bloodmeal in the *CYP4G15*^{Δ0} but not in the *CYP4G15*^{Δ18} homozygous sub-strain. This result indicates a different midgut inducibility of the promoter variants. The manuscript was updated accordingly on lines 135-138.

We also performed an in silico analysis of potential transcription factor binding motifs in the *CYP4G15* promoter sequence. The results are presented in a new Figure S6a. They show that there is no evidence that the Δ18 deletion disrupts a transcription factor

binding motif in the *CYP4G15* promoter. The manuscript was updated accordingly on lines 138-140.

The full *CYP4G15* promoter sequences are provided in Data S2.

Minor comments

- I suggest to change the title as follows: Dengue susceptibility in *Aedes aegypti* is linked to a natural cytochrome P450 promoter variant.

Response: We thank the Reviewer for the suggestion. However, we feel that using the singular form “a natural cytochrome P450 promoter variant” does not adequately reflect the presence of two distinct promoter variants, with and without the $\Delta 18$ deletion. We have amended the title to include the word “promoter”.

- It has been shown that AeCYP4G15 (AAEL006824) is preferentially expressed in mosquito oenocytes (see Martins et al. 2011, doi:10.1590/s0074-02762011000300009). In the present study, the role of this gene in DENV susceptibility is thought to be associated to its transient regulation in the midgut of adult females. How can the reader be sure that the effect of this P450 variant is restricted to the midgut and not more related to other tissues?

Response: To address this interesting point, we have added the following paragraph to the discussion on lines 159-169:

“The specific mode of action through which *CYP4G15* exerts its antiviral effect remains to be investigated. Enzymes of the *CYP4G* subfamily are known to catalyze the synthesis of cuticular hydrocarbons in insects (Qiu et al. 2012; Feyereisen et al. 2020). These hydrocarbons facilitate desiccation resistance, modulate water loss, function as chemical signaling molecules, and play a role in the detoxification of xenobiotics. A previous study detected abundant transcripts of *CYP4G15* in *Ae. aegypti* oenocytes (Martins et al. 2011). Interestingly, our transgenic reporter lines of the *CYP4G15* promoter variants also displayed high levels of GFP expression that predominantly localized within pupal oenocytes (Fig. 3c). Oenocytes are ectodermic cells located in the fat body of insects, including mosquitoes, where they are involved in lipid metabolism and the biosynthesis of cuticular hydrocarbons (Grigoraki et al. 2020). It is possible that *CYP4G15* expression in oenocytes contributes to the antiviral effect observed in the midgut.”

- While getting through RNA-seq methods, I noticed that no minimum ‘normalized read count’ threshold was applied to the dataset, leading to retain genes showing very low

read count across samples (which usually often leads to overestimating fold changes through random draw effect). I understand that applying such filter will likely remove some differentially expressed genes but this may also help consolidating this tissue-specific RNA-seq dataset.

Response: We have re-run the entire transcriptomic analysis with a minimum expression threshold of 10 reads per transcript. This resulted in filtering out 10,309 out of 22,715 genes and all the results have been updated accordingly. The list of statistically significant differentially expressed genes remained largely unchanged after the new analysis. Of note, all datasets have been updated in the GEO repository.

- Two distinct *Ae. aegypti* forms appears to coexist in Central Africa (*aegypti aegypti* and *aegypti formosus*, see Crawford et al. doi.org/10.1101/2024.07.23.604830) but it is not said which form is represented in the Bakoumba strain. In this concern, I suggest providing more information about the origin and the biological nature of Bakoumba strain in the method section. In addition, it would be informative for the reader to know the *aegypti/formosus* status of field collected individuals genotyped for the *cyp4g15* promoter deletion in fig4d.

Response: To determine the subspecies assignment of the Bakoumba strain, we have identified SNPs diagnostic for *Ae. aegypti formosus* (*Aaf*) and *Ae. aegypti aegypti* (*Aaa*), using the whole-genome sequences of individuals from Entebbe (Uganda) and Santarem (Brazil) as reference representatives, respectively (Rose et al. 2020). To isolate the most discriminant markers, 1,798 biallelic SNPs were selected under the criterion of near fixation (frequency >0.95) or near absence (frequency <0.05) in *Aaf*, coupled with the opposite allele frequencies in *Aaa*. Among these diagnostic 1,798 SNPs for differentiation between the two subspecies, 154 were present in the pooled exome-sequencing data available for the Bakoumba strain (Dickson et al. 2020). The genetic background of the Bakoumba strain was determined by examining the allele frequency spectrum of these 154 SNPs. According to the results, shown in a new Figure S9, the Bakoumba strain is predominantly assigned to the *Aaf* subspecies, consistent with genetic ancestry patterns observed in wild-caught specimens from Gabon (Rose et al. 2020).

Line 146: 'is not an artefact of lab colonization'. Not sure to understand this... Although laboratory colonization can lead to strong demographic effects, there is barely no chance that colonization would lead to the de novo emergence of such mutation...

Response: We have rephrased this sentence to clarify its meaning as follows: "This indicates that the *CYP4G15*^{Δ18} variant occurs naturally in wild mosquito populations from West and Central Africa at frequencies similar to those observed in the Bakoumba

strain.”

Line 148: change to ‘a naturally occurring variant’

Response: As noted above, we respectfully disagree with using the singular form “a naturally occurring variant” because we believe it does not adequately reflect the presence of two distinct promoter variants, with and without the $\Delta 18$ deletion.

Lines 151-153: Not sure to understand this. I believe that the low frequency of this variant in natural populations and its relative narrow geographic range may also support a significant fitness cost impairing its spread throughout Africa?

Response: We have removed this statement.

Lines 165-166: I am not sure this example is pertinent considering that *cyp4g15* does not typically show the profile of a xenobiotic-metabolising P450 such as CYP6Ps. Indeed, this gene does not belong to classical P450 subfamilies associated with xenobiotic metabolism in mosquitoes, is highly conserved among species (slow evolution) and is preferentially expressed in oenocytes. Though this gene has been associated to insecticide resistance (Grigoraki et al 2020 Elife, Despres et al Biol letters 2014, Reid et al 2014 J Med Entomol) its role in CHC synthesis is far more credible.

Response: Following the Reviewer’s advice, we have removed the *CYP6P* example. We have also added the following sentences to the discussion on lines 160-163:

“Enzymes of the *CYP4G* subfamily are known to catalyze the synthesis of cuticular hydrocarbons in insects (Qiu et al. 2012; Feyereisen et al. 2020). These hydrocarbons facilitate desiccation resistance, modulate water loss, function as chemical signaling molecules, and play a role in the detoxification of xenobiotics.”

Lines 323-324: Degenerate primers were used for comparing the expression of *cyp4g15* between individuals and pure lines carrying or not the promoter deletion to ‘accommodate for polymorphisms’. But then I wonder how such polymorphisms may affect the *cyp4g15* transcription data presented early on in the manuscript.

Response: We have compared the ability of both primer pairs to quantify *CYP4G15* expression, as a function of the *CYP4G15* promoter genotype. As shown in the figure below, the expression levels are consistent between the primer pairs for the *CYP4G15* ^{$\Delta 18$} variant (left) but not for the *CYP4G15* ^{$\Delta 18$} variant (right). Nevertheless, we consider that any bias in gene expression measurement in our early experiments due to this discrepancy would have been minimal because the frequency of the *CYP4G15* ^{$\Delta 18$} variant in the Bakoumba strain was only ~10% then (Fig. 4a).

Line 357: in Data S2, you may want to indicate the inferred function for novel transcripts.

Response: We have revised this information (now Data S1) to include the functional annotations of novel transcripts when they could be predicted.

Lines 430-431: Such polymorphism dataset may allow investigating if other coding variants are associated with the 18bp promoter deletion...

Response: We have performed a linkage disequilibrium (LD) analysis of the *CYP4G15* locus to determine if the $\Delta 18$ deletion could be associated with SNPs in the protein-coding region of the gene. Using whole-genome sequencing data publicly available for 43 wild-caught *Ae. aegypti* from Gabon (Rose et al. 2020), we found very limited LD between variants in the promoter and protein-coding regions (Fig. S6b). This suggests that the $\Delta 18$ deletion acts independently of SNPs in the protein-coding region of the gene. We added this point in the discussion on lines 176-179.

Fig4d. Please provide allele frequencies instead of occurrence frequencies.

Response: We appreciate the Reviewer's suggestion. However, we believe that displaying the allele frequencies may reduce clarity, as the occurrence frequencies are relatively low in natural populations (<15.4%). Consequently, the allele frequencies would be difficult to discern in pie charts. Moreover, the shallow sequencing depth of publicly available genome sequences often makes it challenging to differentiate between homozygous and heterozygous individuals. As a result, we can estimate the occurrence of the deletion more accurately than its frequency.

- All figures. Please change 'normalized expression' into 'normalized transcript level'.

Response: All the figures were revised according to the Reviewer's suggestion.

- Whole manuscript: the term 'wild-type strain' is confusing. Change to 'field-derived strain'.

Response: We have clarified the term “wild-type” in the Methods section to mean “not genetically modified”, and we choose to avoid using “field-derived”, as all mosquito strains are inherently field-derived.

Reviewer #2 (Remarks to the Author):

This is a novel approach for mapping genes of interest in virus susceptibility – focusing on comparisons between individuals that vary within a population. The authors survey gene expression on days 1 and 2 post feeding in the midgut and find only a few number of genes associated with differences in infection status and that they do not overlap. This suggests distinct mechanisms for controlling infection phenotypes recognizable at different DPI. They follow up with some robust functional work focused on a single cytochrome P450 gene and identify an upstream deletion associated with differences in expression. They also make homozygous lines to look at susceptibility to 2 DENV serotypes and explore the presence of the variant in published sequences. P450's have not traditionally been associated with antiviral activity. This is a nice contribution to the field.

The writing is clear and the experimental approaches robust.

I just have some suggestions that will not require any further experimental work.

What were the full list of 18 genes you found at DPI 1? What are their functions? A summary of these genes should be included (not just relegated to the Supp). Why did you focus on this single gene for this study? Please add the reasoning to the text (couldn't knock down expression of the others?, functions uninteresting?). Wanting to save the other genes for other papers does not seem like a good enough reason as this paper is not very long. They should at least be discussed. Functional work could follow in other papers.

Response: We have provided a justification to the choice of *CYP4G15* as the primary focus of the functional follow-up experiments (lines 87-92). In short, we selected 11 targets for functional testing, prioritizing genes that had a predicted or confirmed link to immunity or metabolism. We have included a new Figure S3 presenting the results of the gene silencing assays for the other 10 shortlisted candidate genes, also listed in Data S1.

Does this variant occur outside Africa? Or does this suggest a single local evolutionary event + spread. Finding it more frequently in a single copy only could indicate fitness issues. I would remove the statement about the variant unlikely causing fitness effects.

Response: The *CYP4G15*^{Δ18} variant is not found outside Africa (Fig. 4d), so it is more likely to have spread locally in Central Africa. The shallow sequencing depth of publicly available genome sequences often makes it challenging to differentiate between homozygous and heterozygous individuals, therefore we cannot reliably detect a deviation from expected heterozygosity. Following the Reviewer's advice, we have removed the statement about fitness effects.

Reviewer #3 (Remarks to the Author):

Merkling et al. investigated variation in DENV infection within an *Aedes aegypti* population by comparing the transcriptomes of DENV-infected and uninfected mosquito midguts at 1 and 2 days post-infection. Among the differentially expressed genes, they focused on a p450 gene, *CYP4G15*. The expression of this gene exhibited variation within the population. When the gene was silenced, the proportion of infected mosquitoes increased, but not virus replication. Conversely, overexpression of the gene resulted in a reduced proportion of DENV-infected mosquitoes using DENV-1 and DENV-3. Comparison at the genomic level revealed variation in the promoter region of *CYP4G15*, with those possessing an 18-bp deletion showing significantly lower *CYP4G15* expression. This deletion was also associated with a higher proportion of infected mosquitoes. The results of this study are very interesting, revealing a mosquito genetic factor involved in variation in their susceptibility to DENV. This is the first time a p450 gene is shown to be involved in resistance to DENV. The manuscript is also well-written. I have only minor comments.

Line 85-86: what other genes were differentially expressed, and why was *CYP4G15* selected among the rest?

Response: We have provided a justification to the choice of *CYP4G15* as the primary focus of the functional follow-up experiments (lines 87-92). In short, we selected 11 targets for functional testing, prioritizing genes that had a predicted or confirmed link to immunity or metabolism. We have included a new Figure S3 presenting the results of the gene silencing assays for the other 10 shortlisted candidate genes, also listed in Data S1.

Line 169: A quick search on PubMed (DOI: 10.4269/ajtmh.18-0607) showed that, in at least one instance, upregulation of *CYP4G15*, among other things, is associated with insecticide resistance in *Aedes aegypti*, which might be worth mentioning.

Response: We have added this reference in the discussion on lines 187-189, as follows:

“A previous study observed that the expression of *CYP4G15* was higher in an insecticide-resistant *Ae. aegypti* strain compared to a susceptible counterpart (Lien et al. 2019)”.

Lines 270-271: providing the g force for the speeds rather than rpm is better.

Response: We have converted rpm to g units.

Line 284, 337: RNase

Response: We have corrected these typos.

Line 360: to create a matrix

Response: We have corrected this typo.

DATA S2 and line 82: there are only 7 genes listed in DATA S2.

Response: We have clarified in the caption that the other differentially expressed genes are shown in separate tabs.

Below are the point-by-point responses (in blue font) to the Reviewer' comments (in black font).

Reviewer #1 (Remarks to the Author):

First, I would like to acknowledge the additional work performed by authors to answer reviewers' comments. Overall, I believe that the additional data provided and the modifications made to the manuscript improved its overall quality, making it suitable for publication in Nature Communications.

Below are two last comments that may help authors to further improve the manuscript.

1) The additional data provided suggest that the Bakoumba strain is a mixture of *Ae. aegypti aegypti* and *Ae. aegypti formosus* with the later genetic background being predominant.

Such finding raises the question of the possible association of the CYP4G15 promoter deletion variant to a specific molecular form, which can be related to its low observed frequency in the field and its narrow geographical distribution in Fig4d. In this view, it would be interesting to know the molecular form of the field caught specimens genotyped for the deletion in Fig4d. In other words, is there any association between the presence of the CYP4G15 deletion variant and a given molecular form?

Similarly, considering that both forms are interfertile but that introgression will lead to chromosomal regions being closer to one or the other molecular form, it would be interesting to compare the genomic sequence surrounding the CYP4G15 locus between 'deletion' and 'wild type' homozygous lines and see if they rather match to one or the other molecular form.

I believe that the possible association of this molecular variant to a particular *Ae. aegypti* molecular form of interest for the reader as it may provide clues about its origin, its dynamics in the field and its association with DENV transmission. In this view I think such questions are worth to be discussed.

Response: The allele frequency distribution inferred from the pooled exome-sequencing data at sites diagnostic for *Aaf* is not suitable for a formal admixture analysis. Therefore, conclusions about the genetic ancestry of the Bakoumba population cannot be drawn from this alone. Instead, we performed an f_3 -statistic analysis to assess whether Bakoumba is admixed between the reference populations from Uganda (*Aaf*) and Brazil (*Aaa*). Our analysis produced an f_3 value of 0.24 with a z-score of 108.23, clearly indicating no evidence of admixture in the Bakoumba population. This finding aligns with

the existing knowledge of the genetic structure of *Ae. aegypti* in Africa, where no admixture signals have been detected in populations from nearby regions in Gabon (La Lopé, Franceville). Consequently, the *CYP4G15* promoter deletion is most likely associated with the *Aaf* ancestry, which is further supported by the absence of the deletion in populations outside Africa.

2) minor comment,

I understand the authors' point of view about not using the singular form 'a naturally occurring variant' though I believe that the term "variant" implies that there are at least two distinct forms coexisting in natural populations.

Response: We appreciate the Reviewer's comment and stand by our choice to use the plural form.